# Population Genetic Analysis in Persimmons (*Diospyros kaki* Thunb.) Based on Genome-Wide Single-Nucleotide Polymorphisms

**DOI:** 10.3390/plants12112097

**Published:** 2023-05-24

**Authors:** Seoyeon Park, Ye-Ok Park, Younghoon Park

**Affiliations:** 1Department of Horticultural Science, Pusan National University, Miryang 50463, Republic of Korea; parksy@jenong.co.kr; 2Sweet Persimmon Research Institute, Gyeongsangnam-do Agricultural Research and Extension Services, Gimhae 50871, Republic of Korea; cjsw98@korea.kr

**Keywords:** *Diospyros* spp., genetic diversity, genotyping-by-sequencing, persimmon, population genetics, single nucleotide polymorphisms

## Abstract

This study investigated the genetic diversity and population structure of a persimmon (*Diospyros kaki* Thunb., 2n = 6x = 90) collection in South Korea by evaluating 9751 genome-wide single-nucleotide polymorphisms (SNPs) detected using genotyping-by-sequencing in 93 cultivars. The results of neighbor-joining clustering, principal component analysis, and STRUCTURE analysis based on SNPs indicated clear separation between cultivar groups (pollination-constant nonastringent (PCNA, 40 cultivars), pollination-constant astringent (PCA, 19), pollination-variant nonastringent (PVNA, 23), and the pollination-variant astringent type (PVA, 9)) based on the astringency types, while separation between PVA and PVNA-type cultivars was unclear. Population genetic diversity based on SNPs showed that the proportions of polymorphic SNPs within each group ranged from 99.01% (PVNA) to 94.08% (PVA), and the PVNA group exhibited the highest genetic diversity (He = 3.86 and uHe = 0.397). *F* (fixation index) values were low ranging from −0.024 (PVA) to 0.176 (PCA) with an average of 0.089, indicating a deficiency of heterozygosity. Analysis of molecular variance (AMOVA) and *F*st among cultivar groups indicated that variation within individuals was higher than that among the groups. Pairwise *F*st values among the groups ranged from 0.01566 (between PVA and PVNA) to 0.09416 (between PCA and PCNA), indicating a low level of cultivar type differentiation. These findings highlight the potential application of biallelic SNPs in population genetics studies of allopolyploids species and provide valuable insights that may have significant implications for breeding and cultivar identification in persimmon.

## 1. Introduction

Most members of the genus *Diospyros* (family: Ebenaceae), which comprises approximately 400 plant species, are distributed in the tropical regions of Asia, Africa, and Central and South America. However, some species, such as Japanese persimmon, are distributed in temperate regions [1,2]. Among them, the following four species are fruit trees: persimmon (*Diospyros kaki* Thunb.), ‘Goyum’ (*Diospyros lotus* L.), American persimmon (*Diospyros virginiana* L.), and ‘Youshi’ (*Diospyros oleifera* Cheng) [2]. These four species are rich sources of nutrients, including minerals, calcium, potassium, vitamins A and C, and carbohydrates [3]. In China, *D. kaki*, *D. lotus*, and *D. oleifera* are cultivated as fruit trees, whereas *D. lotus* and *D*. *oleifera* serve as sources of raw and dried fruits and persimmon oil (tannin), respectively [2]. There is limited knowledge about the origin of *D*. *kaki* and its relationship to other *Diospyros* species. Persimmon cultivation is believed to have begun more than 2000 years ago in China, which is known as a primary center of genetic origin. Then, it spread to Korea, Japan, and several European countries for centuries as an important fruit crop [2]. The world’s total persimmon production was 4,246,319 tons in 2019 with China (3,247,068 tons) as the highest producer followed by Korea (258,874 tons), Spain (245,000 tons), Japan (208,200 tons), Azerbaijan (177,129 tons), and Brazil (167,721 tons) [4].

Polyploidy is commonly observed in fruit tree species [5]. Persimmons exhibit karyotypes of various ploidies. Some cultivars of persimmons are nonaploid (2n = 9x = 135), such as ‘O-tanenashi’ and ‘Hiratanenshi’ [6]. However, most cultivars of *D*. *kaki* are hexaploid (2n = 6x = 90) and are alloploid with complete fertility [2,6]. In contrast, *D. lotus* L. and *D. oleifera* Cheng are diploids (2n = 2x = 30), while *D. virginiana* L exhibits two karyotypes (tetraploidy or hexaploidy) [2]. The size of the draft genome assembly of the diploid persimmons ‘*D*. *oleifera*’ is approximately 812.3 Mb (0.83 pg/1C) with 28,580 genes [7,8]. *D*. *oleifera* is one of the ancestral species of allohexaploid cultivars [8]. Another draft genome was assembled for the diploid ‘Gouyum’ (*Diospyros lotus* L.), a close wild relative of the oriental persimmon [9]. However, a reference genome assembly for hexaploid persimmon has not been reported, which may be due to its genetic complexity.

*D. kaki* Thunb. is classified into the following two cultivar groups based on the presence or absence of an astringent flavor in the fruits during harvest: sweet persimmon and astringent persimmon. Each group can be classified into variant and constant cultivar types depending on the relationship between the presence of seeds and flesh color. The flesh color of the variant type is affected by pollination. If seeds are produced as a result of pollination, the flesh color of the variant type is darker. Conversely, pollination or the presence of seeds does not affect the flesh color of the constant type. Therefore, the variant and constant types are referred to as pollination-variant and pollination-constant, respectively. Based on the effects of seeds on the astringent taste and deintercalation of mature fruits, persimmons can be astringent or nonastringent and are classified into the following four cultivar types: pollination-constant nonastringent (PCNA), pollination-constant astringent (PCA), pollination-variant nonastringent (PVNA), and pollination-variant astringent (PVA) [2,10]. Among the persimmons currently cultivated in South Korea, ‘Fuyu’’ accounts for 83.4% of sweet persimmons [11], while ‘Gapjubaekmok’ accounts for the largest proportion of astringent persimmons. The collection and preservation of South Korean persimmon genetic resources are under the jurisdiction of a government agency (the Sweet Persimmon Research Institute (SPRI) (Changwon, South Korea)). In total, 210 varieties are preserved and used as breeding materials.

The genetic diversity and relationship of persimmons have been studied using molecular markers, such as random amplified polymorphic DNA (RAPD) [12,13], amplified fragment length polymorphism (AFLP) [14,15], simple sequence repeat (SSR) [10,16], start codon-targeted (SCoT) and inter-retrotransposon-amplified polymorphism (IRAP) marker [17]. In these previous studies, a high level of marker polymorphisms and significant inter- and intra-specific genetic diversity were identified in persimmon accessions, which were further characterized by clustering patterns based on geographical region and cultivar type. Recently, the number of genomic analyses has increased with the advent of next-generation sequencing (NGS) technology [18]. Genotyping-by-sequencing (GBS) has the advantage of being relatively inexpensive and rapid [19,20]. In GBS, a large number of single nucleotide polymorphisms (SNPs) can be searched by sequencing only a specific region of the genome using restriction enzymes [21,22]. However, to the best of our knowledge, detailed population genetic analyses using large-scale genome-wide SNP markers and genetic diversity applicable to persimmons have not been performed.

This study aimed to analyze the genetic diversity and population genetics of 93 cultivars at the SPRI based on large-scale SNPs identified using GBS.

## 2. Results

### 2.1. GBS and SNP Discovery

To identify genome-wide SNPs, 93 persimmon cultivars were subjected to GBS analysis (Table 1). The GBS results are summarized in Table 2. The sum of raw reads of each cultivar was approximately 752 million (average per sample: 7.9 million), while the total length of raw reads was 113.5 Gbp (average read length per sample: 1.2 Gbp). The total number of trimmed reads was 712.5 million (average = 7.4 million). The length of all trimmed reads was 92.6 Gbp (average = 1 Gbp). The average length of the trimmed reads was 129.7 bp, while the trimmed reads accounted for 94.70% of the total number of raw reads. Among them, the number of mapped reads was 301.7 million (42.08%) with the average mapped read number being 3.1 million per cultivar. The reference genome coverage ranged from 0.25% to 1.40% with an average of 0.95% per cultivar.

An integrated SNP matrix was prepared using raw SNPs from 93 cultivars. The total number of observed SNPs was 236,947, ranging from 63,916 (‘Hongchoo’) to 9774 (‘Black Sweet Persimmon’), with an average of 44,804 per cultivar. SNPs that passed the filter criteria were classified as homozygous, heterozygous, or other types. The number of homozygous SNPs ranged from 7071 to 42,796 (average: 28,068), while that of heterozygous SNPs ranged from 987 to 5614 (average: 4447). The remaining SNPs could not distinguish between types. After excluding SNPs that had a minor allele frequency (MAF) greater than 5% (49,926) and missing data greater than 30% (95,095), 9751 SNPs were selected (Appendix A). The genome coverage and distribution of the selected SNPs for each chromosome are represented as a heatmap in Figure 1.

### 2.2. Population Structure and Phylogenetic Analysis

Genetic analyses of the population structure, principal component analysis, and hierarchical clustering were conducted for 93 persimmon cultivars comprising four cultivar types (PCA, PCNA, PVA, and PVNA) using 9751 SNPs. Population structure analysis revealed that estimation of the delta K parameter inferred the most suitable model with five subpopulations (K = 5) (Figure 2a). The grouping of cultivars according to cultivar type represented distinct subpopulation structures (Figure 2b). However, several cultivars could not be distinguished based on their subpopulation structures. For example, ‘Black Sweet Persimmon’ and ’09-12-1′ are Chinese PVNA-type and Korean PCNA-type cultivars, respectively, but they exhibited the same subpopulation structure (population 1; pop1). Similarly, ‘Gosho’ of the Japanese PCNA-type cultivar and ‘Wasesaijo’ of the Japanese PCA-type cultivar exhibited the same subpopulation structure (pop2). Additionally, most cultivars had an admixture of at least two different subpopulations. The major subpopulation members in different cultivar types were as follows: PCNA, pop1, pop2, and pop4; PVNA, pop1, pop3, and pop4; PCA, pop1 and pop3; PVA, pop4 and pop5.

Hierarchical clustering using 9751 SNPs was performed to evaluate the genetic relationships among 93 cultivars. Two major clusters (I and II) were observed in the phylogenetic tree (Figure 3). In general, cultivars are divided into independent groups based on their cultivar types. However, some groups comprised cultivars of different types. Of the 93 cultivars, 42 belonged to Cluster I with PCA, PVA, and PVNA being predominant cultivars. Cluster I can be further divided into three subclusters. Subcluster I-1 mainly comprised PCA cultivars, while Subcluster I-2 mainly comprised most PVNA cultivars and some PVA cultivars. In Subcluster I-3, several PVA and PVNA cultivars and one PCNA cultivar (‘Hanagosho’) were grouped. Cultivars of similar origins were closely related to each other in the phylogenetic tree. ‘Cheongdo-Bansi’ (PCA) and ‘Gyeongsan-Bansi’ (PCA) were also similarly related to ‘Haman-Bansi’ (PCA), ‘Sancheong-Danseongsi’ (PCA), and ‘Sangjudungsi’ (PCA), which are all PCA-type cultivars originating from Korea. However, among the Subcluster I-1 cultivars, ‘Hachiya,’ which is a PVA cultivar similar to ‘Yaoki’ (PVA), was grouped into the same group as PCA cultivars. ‘Diamond leaf persimmon,’ whose cultivar type is unknown, belonged to Subcluster 1-1, indicating that it is a PCA-type cultivar. ‘Tonewase’(PVA) and ‘O-tanenashi’ (PVA), which are bud mutations of ‘Hiratanenashi’ (PVA) were closely grouped in Subcluster I-2.

In Cluster II, 51 cultivars were identified with the PCNA-type accounting for most of the cultivars (39 varieties). Cluster II can be divided into two subclusters. In Subcluster II-1, 35 of the 40 varieties were PCNA-type cultivars. ‘Uenishiwasw’ and ‘Sunami’ are bud mutations of ‘Matsumotowase-Fuyu’ and ‘Fuyu’, respectively, and these cultivars were closely located in Subcluster II-1. Similarly to Subcluster I-3, Subcluster II-2 comprised a mixture of 11 cultivars of four different types. The new Korean cultivar ‘Gamnuri’ (PCA), which is a processing cultivar obtained by crossing ‘Sunami’ (PCNA) with ‘Johongsi’ (PVNA), was classified into the same group as ‘Migamjosang’ (PVNA) and ‘12-9-4’ (PVNA). An unknown cultivar type, ‘11-6-120′, was located most closely to ‘Johongsi’ (PVNA) in Subcluster II-2.

The clustering patterns indicated that cultivars originating from the same country tended to be closely grouped. Most Korean PCA-type cultivars were distinct from Japanese PCA-type cultivars. Japanese PVNA-type cultivars were clustered as an independent group from the Korean PVNA-type cultivars. For the PCNA-type, cultivars were closely related in two different countries. However, insufficient sample numbers limited the evaluation of European (two cultivars) and Chinese cultivars (four cultivars).

Principal component analysis of 93 persimmon cultivars based on 9751 filtered SNPs (Figure 4) revealed that the cultivars can be divided into two major clusters. This clustering pattern was consistent with the results of the phylogenetic tree (Figure 3). Most PVNA-type and PCA-type cultivars were grouped in Cluster I, while most PCNA cultivars s were classified in Cluster II. In Cluster II, six PCNA, four PCA, and two PVA cultivars were grouped with PCNA cultivars, whereas one PCNA cultivar was located in Cluster I. As shown in the phylogenetic tree, the PCA group was closer to the PVNA group than to the PCNA group.

### 2.3. Population Genetic Diversity Based on SNPs

Gene diversity analysis of the SNP loci was performed on 91 cultivars except 2 cultivars with unknown astringency types. Based on the maximum number of SNPs allowed as input data for running the GenAlex software (v.6.50), 8095 out of 9751 SNPs were selected by filtering out the SNPs with high heterozygosity rates (Appendix A). In the total population, the number of alleles (Na) was 2.0 for all loci, while the effective number of alleles (Ne) ranged from 1.119 (locus: DLO_r1.0ch15_178451170) to 2.0 (DLO_r1.0ch15_20060874) with an average of 1.711 per locus. Observed heterozygosity (Ho) ranged from 0.0 (20 loci) to 0.765 (DLO_r1.0ch04_4992371) with an average of 0.350. Expected heterozygosity (He) ranged from 0.106 (DLO_r1.0ch15_17845117) to 0.500 (298 loci) with an average of 0.403, while unbiased He (uHe) ranged from 0.107 (DLO_r1.0ch15_17845117) to 0.504 (22 SNPs) with an average of 0.405. Fixation index values were calculated to detect genetic subdivisions and ranged from −0.536 (DLO_r1.0ch14_15726033) to 1.00 (20 SNPs) with an average of 0.169 (Table 3 and Appendix A).

The level of genetic diversity of the four cultivar types as subpopulations was calculated. The proportions of polymorphic SNPs within the PCA (19 cultivars), PCNA (40), PVA (9), and PVNA (23) groups were 98.85%, 97.58%, 94.08%, and 99.01%, respectively. MAF and polymorphic information content (PIC) values were in the range of 0.00−0.50 and 0.00−0.375, respectively, for all subpopulations based on the cultivar type (Figure 5, Appendix A). The distribution patterns of MAF and PIC values were similar among the subpopulations. Most SNPs exhibited PIC values higher than 0.3 in all subpopulations.

The mean Na of each cultivar group ranged from 1.941 (PVA) to 1.990 (PVNA) and was lower than 2.0, indicating that some SNP loci were monomorphic in all accessions for each cultivar group. For genetic diversity, the value of Ho ranged from 0.319 (PCA) to 0.337 (PVA) and was lower than that of He in all cultivar groups, except for PVA, in which He (0.361) was slightly lower than Ho (0.337). However, the PVNA group exhibited the highest genetic diversity (He = 3.86 and uHe = 0.397). *F* (fixation index) values ranged from −0.024 (PVA) to 0.176 (PCA) among the cultivar groups with an average of 0.089, indicating a deficiency of heterozygosity in this persimmon collection (Table 4 and Appendix A).

Genetic variation among the cultivars (individuals) was evaluated by performing an analysis of molecular variance (AMOVA) and calculating the *F*st among cultivar types (groups) (Table 5 and Appendix A). AMOVA revealed that the variation within individuals was higher than that among the groups. Only 3% of the total genetic variation was attributable to differences among groups, whereas 57% of the total genetic variation was attributed to differences within individuals, indicating high genetic diversity within groups. *F*st was 0.026 (*p*-value = 0.001), indicating low genetic variation among groups. These results indicate high genetic differentiation within the population.

Pairwise *F*st values among the groups ranged from 0.01566 to 0.09416, indicating a low level of cultivar type differentiation (Table 6 and Appendix A). The closest relationship was observed between PVA and PVNA types, while the PCA and PCNA types were the most distantly related.

## 3. Discussion

The genetic diversity in persimmon trees is expected to be high because of the complexity of persimmon’s floral biology, which is characterized by diverse sex phenotypes [23]. Various DNA markers, such as SSR, AFLP, and RAPD have been used for gthe enetic analysis of persimmon species [12,13,14,15,16]. However, limited studies have used genome-wide SNP markers for the genetic analysis of persimmon species [24]. In this study, 9751 SNPs were identified in the GBS data of 93 persimmon cultivars in a germplasm collection in South Korea. These SNPs were used to analyze the genetic diversity and population structure of the collection.

The analysis of Korean collections using GBS-based SNPs revealed genetic differentiation not only among all 93 cultivars but also among astringency types and geographical origins (countries). The results of neighbor-joining clustering, principal components analysis, and STRUCTURE analysis indicated a clear separation between the Korean and Japanese cultivar groups and astringency types, especially between the PCNA and non-PCNA cultivars. AMOVA also revealed significant differences among astringency types. However, most molecular variations were detected among cultivars but not among cultivar groups based on astringency types, which indicates that the cultivar types were genetically similar at the molecular level with most polymorphisms being attributed to individual cultivar differences. Similarly, previous studies using SSR, AFLP, and RAPD have demonstrated differences among cultivars and population differentiation according to geographical origin and astringency type. Badness et al. [25] used RAPDs to evaluate persimmon germplasm resources in Spain and suggested that RAPD fingerprinting data were consistent with the hypothetical origins, adaptation history, and previous classification of persimmons. Guo and Luo [26] evaluated Japanese persimmon cultivars using SSR. Yonemori et al. [14] and Naval et al. [27] used AFLP and microsatellites, respectively, and reported a clear separation between European and Asian cultivars and distinct clustering among Japanese, Chinese, and Korean cultivars. Jing et al. [28] used sequence-related amplified polymorphism markers and reported a clear differentiation at the inter-species and intra-species levels of the persimmon genus.

Genetic variation within a population of persimmons has also been demonstrated in other studies. Guan et al. [29] used 143 alleles obtained from 12 SSR markers and demonstrated frequent intraspecific gene exchange in persimmons. Additionally, the authors reported that the diversity within groups was higher than that between groups in persimmons. Parfitt et al. [15] performed neighbor-joining clustering, multidimensional scaling, and STRUCTURE analysis based on the AFLP markers and reported that the separation of Chinese, Korean, and Japanese cultivar groups and pollination type was weak and not genetically significant. Genetic diversity analysis and AMOVA of European collections (71 Japanese, Italian, and Spanish persimmon cultivars) using 19 microsatellite markers revealed significant genetic variability between astringent-type and country-origin groups (73.3%) but an increased genetic variability within the groups (85.2%) [27].

The results of population genetic analysis based on SNPs in this study were similar to those of previous studies using various types of molecular markers. This indicates the reliability of using the biallelic SNP marker type in genetic studies on allopolyploids. The accuracy of genotyping in polyploids is lower than that in diploids when biallelic SNP genotyping techniques such as GBS are used due to their multi-locus alleles and multiallelic states at a single locus [30,31]. This is further complicated when allelic dosage and allelic variation between subgenomes that co-exist in alloploids such as the persimmon are evaluated without reference genomes [32,33]. For diploid and autoploids, more informative SNP variants can be identified by discriminating sequences of paralogous gene copy, as well as increasing genome coverage and read depth to maximize the recovery of variants. Targeting single-dose SNPs also enhanced the genotyping accuracy in an autotetraploid such as that in [34,35]. The major factor decreasing the SNP genotyping error rate in allopolyploid species is the distinction of non-allelic variants. This is made possible by reducing genome complexity to the diploid level through a subgenome-specific sequencing method, which requires whole genome sequences of each diploid donor taxa. Various studies have reported the genetic diversity and population structure based on genome-wide SNPs for several allopolyploid species including wheat [36,37,38], canola [39], and coffee [40], in which reference sequence data for the subgenome are available.

In conclusion, the study highlights the potential of genome-wide SNP analysis in evaluating genetic diversity and population structure of persimmon cultivars. The outcomes indicate a substantial degree of genetic diversity in the sample collection, with notable differentiation observed among astringency types and geographical regions. Nevertheless, the study finds that most of the genetic variation can be attributed to individual cultivars, suggesting a high level of genetic similarity among cultivar types at the molecular level. These findings are consistent with earlier studies that used different molecular markers, which confirms the robustness of SNP markers in genetic studies on allopolyploid species such as persimmons. The study provides valuable insights into the genetic diversity of persimmon cultivars, which could have significant implications for breeding and cultivar identification.

## 4. Materials and Methods

### 4.1. Plant Materials and Genomic DNA Extraction

In total, 93 persimmon cultivars (40 PCNA, 19 PCA, 23 PVNA, 9 PVA, and 2 unknown) maintained in the Persimmon Research Center, GNARC (Changwon, Republic of Korea), were used as an accession panel for population genetic analysis based on SNPs. For EST-SSR marker development, a sample panel comprising 8 cultivars, including the diploid cultivar ‘Goyum’ (*Diospyros lotus* L) (2n = 2x = 30), was used (Table 1). Genomic DNA was then extracted. Young leaf samples were collected from each cultivar from late April to early May 2019, immediately stored in a −70 °C freezer, and crushed with liquid nitrogen whenever necessary. DNA was isolated from 93 varieties for GBS library preparation and EST-SSR analysis. The leaf sample (1 g) was grounded with liquid nitrogen, and the DNA was extracted using a plant genomic DNA isolation kit (Davinchi-k, Seoul, Republic of Korea). The concentration and purity of the extracted DNA samples were analyzed using a spectrophotometer (Pico 200, Picodrop, Hinxton, UK) and agarose gel electrophoresis using a 1% gel.

### 4.2. GBS and SNP Detection

The genomic DNA (250 ng; 50 ng·µL^−1^) of each sample was used to prepare a set of *Pst*I-*Msp*I sequencing libraries according to the standard GBS library analysis method [21].

The genomic DNA of each of the 93 samples was double-digested with *Pst*I and *Msp*I (New England Biolabs, Ipswich, MA, USA) and 200 U T4 DNA ligase was added to the restriction enzyme-treated fragments to attach the adapter-containing barcode and a common adapter, after which pooling was carried out for preparing 96 samples. After pooling, the suitability of the constructed libraries for GBS analysis was evaluated using gel electrophoresis. The prepared library was purified using the Qiagen MinElute column and sequenced using the paired-end sequencing method with Hiseq 2000. Raw GBS data were demultiplexed using the barcode sequences, and the barcode and adapter sequence were removed using the Cutadapt program (v.1.8.3) [41]. Then, sequence quality trimming to refine cleaned reads was performed using the DynamicTrim program and Lengthsort program of the SolexaQA (v.1.13) package [42]. DynamicTrim removed low-quality bases with a Phred score of < 20 at both ends of the short read and Lengthsor filtered for a short read with a ≥25 bp length. The cleaned reads were mapped to the persimmon reference genome assembly (Persimmon Database, http://persimmon.kazusa.or.jp, accessed on 10 May 2021) using the BWA (0.6.1-r104) program [43], and a BAM format file was created. Raw SNPs were detected from the generated BAM file using the SAMtools program (0.1.16) [42] under the criteria of the minimum mapping quality of SNPs (-Q) = 30 and minimum read depth (-d) = 3, and consensus sequences were extracted. The final SNP matrix was prepared after filtering the miscalling SNP loci through SNP comparison between samples by using the *SEEDERS in-house* script [44]. The SNPs were classified as follows:-Homozygous: SNP read depth ≥ 90%-Heterozygous: 40% ≤ SNP read depth ≤ 60%-Others: 20% ≤ read rate < 40% or 60% < read rate < 90%.

Finally, SNP filtering was performed to select SNPs at a level suitable for subsequent analysis based on biallelic SNP loci, MAF of > 5%, and missing rate of < 30% by using the *SEEDERS in-house* script. The genomic distribution of filtered SNPs was visualized using the basic plotting function of the Circos program [45] in R version 2.1.5.2 (http://www.r-project.org/, accessed on 17 May 2021).

### 4.3. Population Genetics Analysis

#### 4.3.1. Population Genetic Diversity

Genetic diversity analysis was performed using GenAlex program (v.6.50) [46] with the following parameters: Na, number of effective alleles (Ne), Ho (Ho = number of heterozygosity/N), He (He = 1 − Σ *p_i_*^2^), Shannon’s information index (I = −1 Σ (*p_i_* × ln (*p_i_*)), uHe (uHe = (2N/(2N − 1) × He)), and fixation index (F = (He − Ho)/He), where N is the total number of individuals, *pi* is the frequency of the *i*th allele for the population, and Σ *p_i_*^2^ is the sum of the squared population allele frequencies. The *PIC* value was calculated using the following formula [47]:(1)$$PIC=1-\overset{n}{\underset{i=1}{\sum }}p_{i}{}^{2}-\overset{n-1}{\underset{i=1}{\sum }}\;\overset{n}{\underset{j=i+1}{\sum }}2p_{i}{}^{2}p_{j}{}^{2}$$
where *n* is number of marker alleles and *p_i_* and *p_j_* are the allele frequency in the populations *i* and *j*, respectively. The level and source of molecular genetic variation were analyzed using AMOVA with the GenAlex program.

#### 4.3.2. Population Structure and Phylogenetic Analysis

The population structure of the 93 accessions was analyzed with the admixture-based model (Markov chain Monte Carlo-based Bayesian model) using the STRUCTURE program (v.2.3.4) [48,49]. The range for the K values was set to 1–10. For each K value, the analysis was repeated 10 times by setting a burn-in period of 10,000 iterations. The optimal number of subpopulations (subpops) was determined based on the appropriate K value calculated by the delta-K method [49]. Principal component analysis was performed using the SNPRelate R package [50]. A hierarchical clustering tree was constructed based on Nei’s genetic distance using the Poppr R package [51]. Bootstrap values for the trees were determined based on 1000 replications.

## Figures and Tables

**Figure 1 plants-12-02097-f001:**
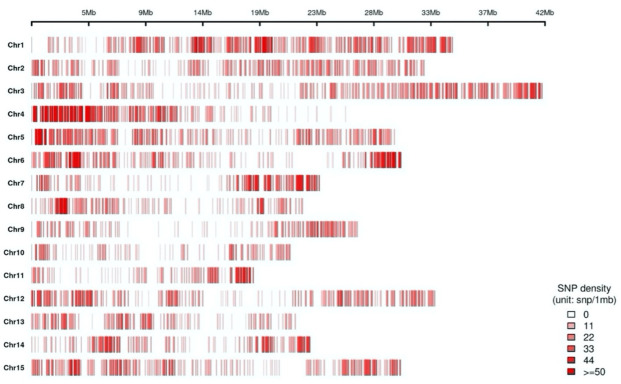
A heat map showing genome coverage and distribution of 9751 single-nucleotide polymorphisms (SNPs) identified using genotyping-by-sequencing of 93 persimmon cultivars (*Diospyros kaki* Thunb). The chromosomal locations of the SNPs (red vertical lines) were determined based on the draft whole genome sequence database of Caucasian persimmon (*D. lotus* L.) (http://persimmon.kazusa.or.jp/, accessed on 17 May 2021) (file name: DLO_r1.0.pseudomolecule.fasta).

**Figure 2 plants-12-02097-f002:**
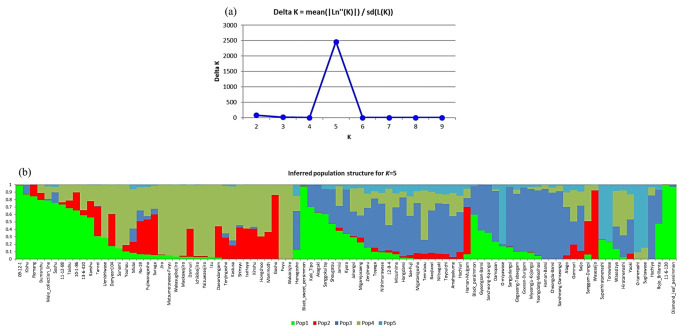
Population structure of 93 persimmon accessions (*Diospyros kaki* Thunb.) analyzed using 9751 single-nucleotide polymorphisms (SNPs). (**a**) The optimal K value determined using Structure Harvester was 5, which indicates that the population can be grouped into five subpopulations. (**b**) The population structure (K = 5) of 93 persimmon cultivars was analyzed using 9751 SNPs based on the admixture-based clustering model of STRUCTURE. The *y*-axis represents the subpopulation members, and the vertical line on the *x*-axis represents the cultivars. Each color indicates a different subpopulation inferred based on the estimation of the delta K parameter. The colored segments within each bar reflect the proportional contributions of each subpopulation to the cultivar.

**Figure 3 plants-12-02097-f003:**
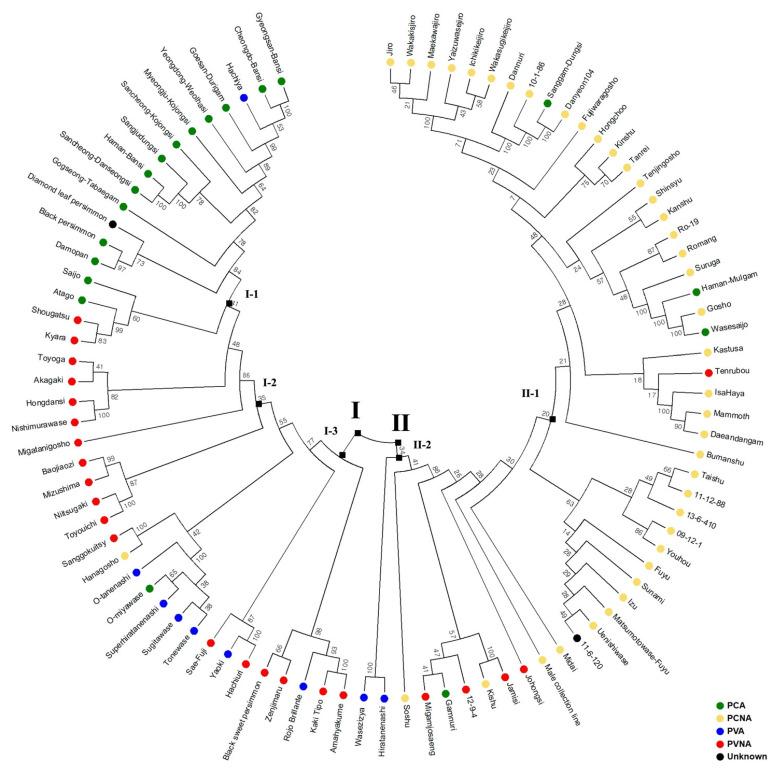
A phylogenetic tree constructed using the neighbor-joining method for 93 persimmon accessions with 9751 single-nucleotide polymorphisms. The pairwise genetic distance was calculated based on Nei’s genetic distance using the Poppr R package. Bootstrap values (as percentages) are shown at each internal node. PCNA: pollination-constant nonastringent; PCA: pollination-constant astringent; PVNA: pollination-variant nonastringent; PVA: pollination-variant astringent.

**Figure 4 plants-12-02097-f004:**
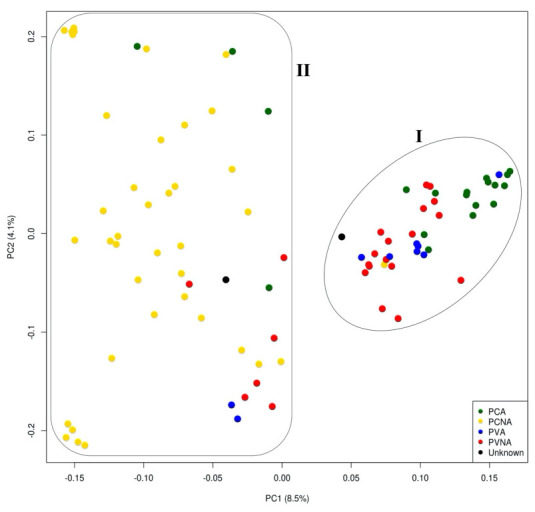
Two-dimensional plot of the principal component analysis results of 93 persimmon accessions (*Diospyros kaki* Thunb.) calculated based on 9751 single nucleotide polymorphisms using the SNPRelate R package. PCNA: pollination-constant nonastringent; PCA: pollination-constant astringent; PVNA: pollination-variant nonastringent; PVA: pollination-variant astringent.

**Figure 5 plants-12-02097-f005:**
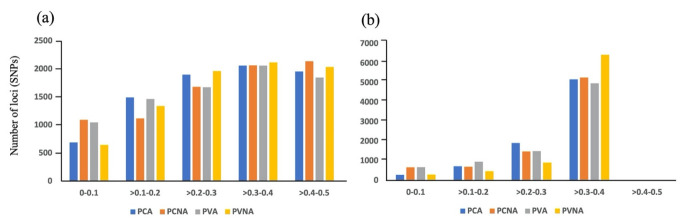
Distribution of the number of loci (single-nucleotide polymorphisms) based on their minor allele frequency (MAF) (%) (**a**) and polymorphic information content (PIC) value (**b**).

**Table 1 plants-12-02097-t001:** List of 93 persimmon accessions (*Diospyros kaki* Thunb.) used in this study.

No.	Cultivar	Type ^1^	Origin
1	Akagaki	PVNA	Japan
2	Amahyakume	PVNA	Japan
3	Atago	PCA	Japan
4	Baojiaozi	PVNA	China
5	Black persimmon	PCA	China
6	Black sweet persimmon	PVNA	China
7	Bumanshu	PCNA	Korea
8	Cheongdo-Bansi	PCA	Korea
9	Daeandangam	PCNA	Korea
10	Damopan	PCA	Japan
11	Dannuri	PCNA	Korea
12	Danyeon104	PCNA	Korea
13	Diamond leaf persimmon	Unknown	China
14	Fujiwaragosho	PCNA	Japan
15	Fuyu	PCNA	Japan
16	Gamnuri	PCA	Korea
17	Goesan-Durigam	PCA	Korea
18	Gogseong-Tabaegam	PCA	Korea
19	Gosho	PCNA	Japan
20	Gyeongsan-Bansi	PCA	Korea
21	Hachiuri	PVNA	Japan
22	Hachiya	PVA	Japan
23	Haman-Bansi	PCA	Korea
24	Haman-Mulgam	PCA	Korea
25	Hanagosho	PCNA	Japan
26	Hiratanenashi	PVA	Japan
27	Hongchoo	PCNA	Korea
28	Hongdansi	PVNA	Korea
29	Ichikikeijiro	PCNA	Japan
30	IsaHaya	PCNA	Japan
31	Izu	PCNA	Japan
32	Jamisi	PVNA	Korea
33	Jiro	PCNA	Japan
34	Johongsi	PVNA	Japan
35	Kaki Tipo	PVNA	Italy
36	Kanshu	PCNA	Japan
37	Kastusa	PCNA	Japan
38	Kinshu	PCNA	Japan
39	Kinshu-2	PCNA	Japan
40	Kyara	PVNA	Japan
41	Maekawajiro	PCNA	Japan
42	Male collection line	PCNA	Korea
43	Mammoth	PCNA	Japan
44	Matsumotowase-Fuyu	PCNA	Japan
45	Midai	PCNA	Japan
46	Migamjosang	PVNA	Korea
47	Migatanigosho	PVNA	Japan
48	Mizushima	PVNA	Japan
49	Myeongju-Kojongsi	PCA	Korea
50	Niitsugaki	PVNA	Japan
51	Nishimurawase	PVNA	Japan
52	O-miyawase	PCA	Korea
53	O-tanenashi	PVA	Japan
54	Ro-19	PCNA	Japan
55	Rojo Brillante	PVA	Spain
56	Romang	PCNA	Korea
57	Sae-Fuji	PVNA	Japan
58	Saijo	PCA	Korea
59	Sancheong-Danseongsi	PCA	Korea
60	Sancheong-Kojongsi	PCA	Korea
61	Sanggam-Dungsi	PCA	Korea
62	Sanggokuitsy	PVNA	Japan
63	Sangjudungsi	PCA	Korea
64	Shinsyu	PCNA	Japan
65	Shougatsu	PVNA	Japan
66	Soshu	PCNA	Japan
67	Sugitawase	PVA	Japan
68	Sunami	PCNA	Japan
69	Superhiratanenashi	PVA	Japan
70	Suruga	PCNA	Japan
71	Taishu	PCNA	Japan
72	Tanrei	PCNA	Japan
73	Tenjingosho	PCNA	Japan
74	Tenrubou	PVNA	Japan
75	Tonewase	PVA	Japan
76	Toyoga	PVNA	Japan
77	Toyouichi	PVNA	Japan
78	Uenishiwase	PCNA	Japan
79	Wakakisjiro	PCNA	Japan
80	Wakasugikeijiro	PCNA	Japan
81	Wasesaijo	PCA	Japan
82	Wasezizya	PVA	Japan
83	Yaizuwasejiro	PCNA	Japan
84	Yaoki	PVA	Japan
85	Yeongdong-Weolhasi	PCA	Korea
86	Youhou	PCNA	Japan
87	Zenjimaru	PVNA	Japan
88	09-12-1	PCNA	Korea
89	10-1-86	PCNA	Korea
90	11-12-88	PCNA	Korea
91	11-6-120	Unknown	Korea
92	12-9-4	PVNA	Korea
93	13-6-410	PCNA	Korea

^1^ PCNA: pollination-constant nonastringent; PCA: pollination-constant astringent; PVNA: pollination-variant nonastringent; PVA: pollination-variant astringent.

**Table 2 plants-12-02097-t002:** Summary of the genotyping-by-sequencing results of 93 persimmon accessions (*Diospyros kaki* Thunb).

Category	Total	Average per Cultivar
No. of total raw reads	751,086,680	7,823,820
No. of trimmed reads	712,437,918	7,421,228
Total length of raw reads (bp)	113,414,088,680	1,181,396,757
Total length of trimmed reads (bp)	92,565,980,293	964,228,961
No. of mapped reads	301,735,646	3,143,080
Total length of mapped regions (bp)	379,067,691	3,948,622

**Table 3 plants-12-02097-t003:** The number of alleles (Na) and the number of effective alleles (Ne = 1/(Σ*pi*^2^)). Shannon’s information index (I = −1 × Σ (*pi* × ln (*pi*)), observed heterozygosity (Ho = number of Hets/N), expected heterozygosity (He = 1 − Σ*pi*^2^), unbiased expected heterozygosity (uHe = (2N/(2N − 1) x He), and fixation index (F = (He − Ho)/He), where pi is the frequency of the ith allele for the population and Σ*pi*^2^ is the sum of the squared population allele frequencies.

Population	Category	N	Na	Ne	I	Ho	He	uHe	F
Total	Max	93	2.000	2.000	0.693	0.765	0.500	0.504	1.00
	Min	66	2.000	1.119	0.217	0.000	0.106	0.107	−0.536
	Mean	78.904	2.000	1.711	0.588	0.350	0.403	0.405	0.169
	SE	0.083	0.000	0.003	0.001	0.002	0.001	0.001	0.003

**Table 4 plants-12-02097-t004:** Mean and standard error (SE) of genetic parameters for each subpopulation of the 91 persimmon accessions.

Population		N	Na	Ne	I	Ho	He	uHe	F
PCA	Mean	16.617	1.989	1.666	0.559	0.319	0.380	0.392	0.176
	SE	0.023	0.001	0.003	0.002	0.002	0.001	0.001	0.004
PCNA	Mean	34.393	1.976	1.661	0.546	0.368	0.373	0.378	0.062
	SE	0.042	0.002	0.003	0.002	0.002	0.002	0.002	0.004
PVA	Mean	8.039	1.941	1.634	0.531	0.377	0.361	0.386	−0.024
	SE	0.013	0.003	0.003	0.002	0.003	0.002	0.002	0.005
PVNA	Mean	19.347	1.990	1.680	0.566	0.339	0.386	0.397	0.145
	SE	0.026	0.001	0.003	0.002	0.002	0.001	0.001	0.004

**Table 5 plants-12-02097-t005:** Results of the analysis of molecular variation (AMOVA) for 4 cultivar types (groups) of 91 persimmon cultivars (individuals) in Korea.

Source	df	Sum of Squares	Estimated Variation	Percent of Variation	Fixation Index (*F*st)
Among groups	3	16221.286	57.668	3%	0.026
Among individuals	87	260542.626	873.294	40%	
Within individuals	91	113582.000	1248.154	57%	
Total	181	390345.912	2179.117	100%	

**Table 6 plants-12-02097-t006:** Results of the analysis of pairwise fixation index (*F*st) values for 4 cultivar types of 91 persimmon cultivars in Korea. The *F*st values and their *p*-values are shown in the lower and upper parts of the table, respectively.

	PCNA	PVNA	PCA	PVA
PCNA		0.0000	0.0000	0.0000
PVNA	0.07184		0.0000	0.01074
PCA	0.09416	0.02762		0.00098
PVA	0.08467	0.01566	0.03575	

PCNA: pollination-constant nonastringent; PCA: pollination-constant astringent; PVNA: pollination-variant nonastringent; PVA: pollination-variant astringent.

## Data Availability

Not applicable.

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
