# Peer review of "Population Genetic Analysis in Persimmons (Diospyros kaki Thunb.) Based on Genome-Wide Single-Nucleotide Polymorphisms"

_plants, 2023, doi:10.3390/plants12112097_

Round 1

Reviewer 1 Report

Dear colleagues,

I enjoyed reading your manuscript. It is simple, strait-forward, and makes a clear case regarding the applicability of the GBS approach in future research of persimmon cultivars. Please read my specific comments below.

L 82: please remove „based on NGS“

I am skipping the Results and Discussion section for now and resuming with the Materials and methods.

L 320 and 327: in the end, how many samples were included in the analysis, 93 or 94? (from the rest of the manuscript, it is obvious 93 individuals were analyzed)

Subsection 4.2. For someone familiar with the GBS approach, everything is clear and nothing should be changed. However, for others, this is not the case. I think you should go into more detail here. Also, references for used tools are needed (Cutadapt, DynamicTrim, SAMtools, etc.).

L334-335 – why did you prepare two sets of libraries?

L 335 – what is the origin of the “standard GBS library analysis method”? Some reference is needed. Imagine someone who wants to repeat your experiment. Do you think this would be possible based on provided information?

L 337 – what quality control analyses?

L347-348 – which software was used?

L353 – how did you perform SNP filtering?

L374 – and how deltaK was reached?

Discussion section

L 262 and 265: Two consecutive sentences start with “In this study…”

Reviewer 2 Report

In the introduction add more significant and recent reports to the story on 

genetic diversity and relationship of persimmons.

in the discussion 

Add more stories about the GBS-based SNPs that reveal diversity in crops under study or related species 

add more literature and both pros and cons on the discussion about the studies that have reported the genetic diversity based on SNPs for several autopolyploid and allopolyploid species with specific mention of the crop under study or related species.

Reviewer 3 Report

This study investigated the genetic diversity and population structure of a collection of South Korean persimmons (Diospyros kaki Thunb., 2n=6x=90). This species is one of the less studied species in molecular research. Genotyping sequencing was used to detect 9751 genome-wide SNPs in the 93 cultivars studied. The varieties were classified into 4 groups based on 2 traits and the degree of separation between the groups was examined using the markers used.  Within each group, a high degree of divergence was detected, while between groups it was not. Genetic diversity of the population based on SNPs showed that the proportion of polymorphic SNPs within each group was high, with the PVNA group showing the highest diversity. Fixation index values were low, indicating a lack of heterozygosity. Variation between individuals was higher than between groups. Paired Fst values between groups were low, indicating low levels of differentiation of cultivar types.

Considering that little is known about the genetic background of Diospyros kaki (reference genome is not yet available), the study under review provides important additional data.

The manuscript is well-edited and well-structured, with all relevant data and analysis, including supplementary material.

General comments:

The introduction should provide more detail on the results of the analysis of persimmon and related species in the same genus using molecular markers. The putative domestication of the cultivated species could also be discussed.

Pay attention to when to use subscripts and superscripts in formulas, as this can lead to misunderstandings.

Round 2

Reviewer 3 Report

All the authors have corrected all the errors indicated and the manuscript has been completed accordingly, and I, therefore, recommend that the article be accepted.